# SCORE-BASED MEMBERSHIP INFERENCE ON DIFFUSION MODELS

## ABSTRACT

Membership inference attacks (MIAs) against diffusion models have emerged as a pressing privacy concern, as these models may inadvertently reveal whether a given sample was part of their training set. We present a theoretical and empirical study of score-based MIAs, focusing on the predicted noise vectors that diffusion models learn to approximate. We show that the expected denoiser output points toward a kernel-weighted local mean of nearby training samples, such that its norm encodes proximity to the training set and thereby reveals membership. Building on this observation, we propose **SimA**, a single-query attack that provides a principled, efficient alternative to existing multi-query methods. SimA achieves consistently strong performance across variants of DDPM, Latent Diffusion Model (LDM). Notably, we find that Latent Diffusion Models are surprisingly less vulnerable than pixel-space models, due to the strong information bottleneck imposed by their latent auto-encoder. We further investigate this by differing the regularization hyperparameters ($\beta$ in $\beta$-VAE) in latent channel and suggest a strategy to make LDM training more robust to MIA. Our results solidify the theory of score-based MIAs, while highlighting that Latent Diffusion class of methods requires better understanding of inversion for VAE, and not simply inversion of the Diffusion process

## 1 INTRODUCTION

Generative image models leave evidence of their specific training data at deployment time in their generative process. While making draws from approximations of $p(x)$ or $p(x|y)$, they leave biases of the training samples (finite and fixed realizations from the real $p(x)$ or $p(x|y)$). These biases may be used in theory to reconstruct the training data, a process known as model inversion (Zhu et al., 2016; Creswell & Bharath, 2018; Carlini et al., 2023; Somepalli et al., 2023a;b; Gu et al., 2023).

The ability to invert these models raises concerns in privacy and intellectual property spaces for specific use-cases of generative models, but also possibly provides unique perspectives into the idiosyncrasies of the generative models themselves. If the models were perfect, they would sample from a distribution indistinguishable from the data generating process; the ways in which they deviate from this distribution inform upon their structure.

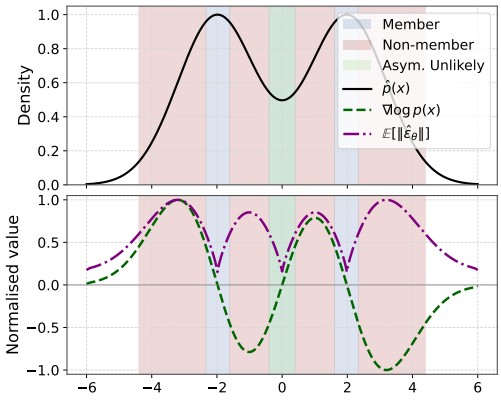

Figure 1: A diagram of our Membership Inference method in one dimension. In blue are regions of high membership likelihood, corresponding to low $\|\hat{\varepsilon}_\theta\|$, plotted in purple. The green region is unlikely to be sampled in high dimensions (c.f. Sec. 3).

A critical precursor to model inversion is the *membership inference attack* (MIA), which determines whether a given image was included in the training set. MIA effectively constructs a classifier for identifying training examples, setting aside the problem of searching the domain for high-likelihood

examples. While this may be the easier sub-problem of full model inversion, it is no less important, as without successful MIA, model inversion is impossible.

Building upon analytic results from the literature about diffusion models (Pidstrigach, 2022; Karras et al., 2022), we describe a performant MIA method that we believe simplifies other current methods into a consistent methodology: the norm of the estimated score at the test point across diffusion times (the $t$ index). Our empirical results show this simple method has AUC scores at or above the previous state-of-the-art for DDPM-weightsets (Ho et al., 2020) on common smaller datasets, and well above the previous state-of-the-art for the related Guided Diffusion model (Dhariwal & Nichol, 2021) on ImageNet-1k.

In contrast to this, our experiments also show that Latent Diffusion Models (LDM) may be robust to MIA attacks of this general class; not only our own method but *every* method tested has decreased LDM performance. Running otherwise identical membership-inference attacks on publicly well-trained DDPM and LDM checkpoints using the same *member* and *held-out* splits, we find dramatic performance drops across all metrics and across all methods. We hypothesize that LDM robustness may be unrelated to the Diffusion part of the Latent Diffusion Model.

We further experiment by training new LDMs on new VAE encoders with differing regularization hyperparameters (the $\beta$ KL-weight in the $\beta$-VAE framework (Burgess et al., 2018)) for the VAE pre-training. We find that MIA performance is susceptible to the KL-weight. We also vary the usage of an additional discriminator between the input image and reconstructed image during VAE training, as suggested by VQ-GAN (Esser et al., 2021). Our modifications show that we can at once reduce the vulnerability of LDMs to MIA while improving generation fidelity as measured by FID. This indicates that successful membership inference and model inversion on the Latent Diffusion class of methods requires better understanding of inversion for VAE, and not simply inversion of the Diffusion process.

In summary, our contributions are the following:

1. A derivation of a simple Membership Inference Attack method (**SimA**), which is a reduction of other methods into a general framework.

2. An empirical demonstration that **SimA** provides top performance on standard datasets and independently trained base models.

3. An empirical demonstration that this general class of methods seems to fail on Latent Diffusion Models, for reasons possibly unrelated to the diffusion generative process itself, indicating a gap in the literature.

All data splits, model checkpoints, training/fine-tuning scripts, and testing code are released on our GitHub repository `https://github.com/mx-ethan-rao/SimA`

## 2 BACKGROUND AND RELATED WORK

**Diffusion Models:** Our membership inference work is specific to diffusion-based generative image models. Originally introduced as score-based generative models (without the explicit connection to the Diffusion model) in Song & Ermon (2019), a very large number of publications have explored variations of these models since that point (Song & Ermon, 2019; Ho et al., 2020; Song et al., 2020; Nichol & Dhariwal, 2021; Dhariwal & Nichol, 2021; Rombach et al., 2022).

While each contribution has its own particular training paradigm and architecture, our attack applies to the broad class of models that estimate a gradient flow field $\hat{\varepsilon}(x, t)$ at points $x$ for a smoothing parameter/diffusion time $t$ that approximates the gradient of the smoothed log-likelihood, $\nabla \log(p(x) * \mathcal{K}(t))$ (Kamb & Ganguli (2024), or Appendix A.4), which is often induced by a conceptual and/or training-time "forward" noise process $x_t = \sqrt{\bar{\alpha}_t} x_0 + \sigma_t \varepsilon$ and a "backward" denoising process similar to denoising auto-encoder processes (Alain & Bengio, 2014). These may be variance-preserving or variance-exploding (Song et al., 2020), based on the exact parameterization of the noise schedule. In this work we directly use weights from the following models: DDPM (Ho et al., 2020), Guided Diffusion (Dhariwal & Nichol, 2021), Latent Diffusion Model (LDM) (Esser et al., 2021), and Stable Diffusion (Rombach et al., 2022). Each of these directly estimate the noise

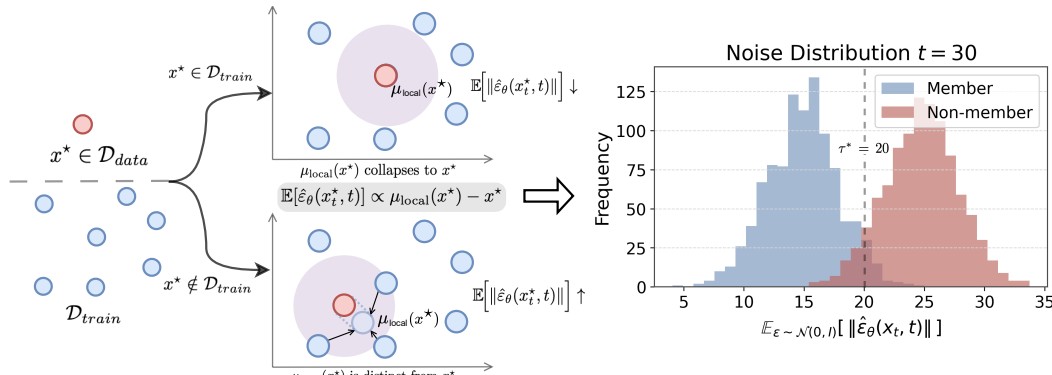

Figure 2: **Score-based MIA intuition with local-mean geometry.** In a small neighborhood ("local ball") around a query $x^\star$, let $\mu_{\text{local}}(x^\star)$ be the kernel-weighted mean of nearby training samples. The model's predicted noise (score) points from $x^\star$ toward this local mean, $\mathbb{E}[\hat{\varepsilon}_\theta(x_t^\star, t)] \propto \mu_{\text{local}}(x^\star) - x^\star$. For members ($x^\star \in \mathcal{D}_{train}$), the local mean $\mu_{\text{local}}(x^\star)$ collapses to training sample $x^\star$, producing small norms, whereas for non-members ($x^\star \notin \mathcal{D}_{\text{train}}$), $\mu_{\text{local}}(x^\star)$ deviates from $x^\star$, yielding larger norms. Right: the histogram at $t = 30$ shows the separation in $\mathbb{E}_{\varepsilon \sim \mathcal{N}(0,I)}[\|\hat{\varepsilon}_\theta(x_t, t)\|]$.

vector $\varepsilon$ using a neural network $\hat{\varepsilon}(x, t)$. Our method manipulates $x$ and $t$ and analyze the networks' outputs, but otherwise is agnostic to the exact architecture and weights.

Latent Diffusion Models (Esser et al., 2021; Rombach et al., 2022) perform their forward process on the latent space of some encoder-decoder structure (usually a Variational Auto-encoder (Kingma et al., 2013)). As we show, this encoder-decoder structure appears to be robust to membership inference, even though the particular diffusion model on its own may not be.

The analytic tractability of these models permits theory about their behavior (Kamb & Ganguli, 2024; Lukoianov et al., 2025), even pre-dating the publication of the DDPM, e.g., Alain & Bengio (2014). Our method is conceptually linked to results and intuition from Alain & Bengio (2014) and Kamb & Ganguli (2024).

**Membership Inference Attack threat models:** Model inversion and membership inference attacks pre-date the introduction of the DDPM and LDM generative image models; MIA was defined originally for general classification tasks (Shokri et al., 2017). A non-trivial amount of literature since that point has focused on MIA for generative image models (Chen et al., 2020) starting with GANs, due to its attack surface (pixel arrays) and its clear privacy and intellectual property implications.

Much of the terminology and structure were defined in the security context, where a threat model defines the scope and allowable resources for an attack vector. So-called "black box" attacks from the original MIA literature are performed without knowledge of the model weights or structure, and can only rely on input and output pairs from a deployed model. In contrast, "white box" attacks (Pang et al., 2023) have access to the full architecture/weight set. We consider the most common class of diffusion model MIAs, "grey-box" attacks (Duan et al., 2023; Zhai et al., 2024; Kong et al., 2023; Matsumoto et al., 2023; Carlini et al., 2023; Fu et al., 2023), which span a range of options between those two extrema; in general they have access to weights and/or internal representations, and may query the model for particular test points. We review each of these grey box attacks in detail in Section 3 and compare to each, with the exception of Zhai et al. (2024) as it performs membership inference on conditional generative models instead of the unconditioned case.

## 3 METHODOLOGY

### METHOD OVERVIEW

The predicted noise $\hat{\varepsilon}_\theta$ is outputted by the neural network, which is a scaled estimator of the score $-\sigma_t \nabla_{x_t} \log p_t(x_t)$ (Song et al., 2020; Ho et al., 2020; Luo, 2022), for data generating distribution $p(x)$ and its mollifications $p_t(x)$ on a noising schedule $\sigma_t$. Our Simple Attack (**SimA**) method is

$$\mathcal{A}(x, t) = \|\hat{\varepsilon}_\theta(x, t)\|. \tag{1}$$

Here, $x$ is the test point, which ostensibly was drawn from $p(x)$, but is either in the training data or not, and $t$ is the diffusion time parameter. We provide a more rigorous derivation and its connections to other MIA methods in the following sections, but we feel that its intuition is also instructive on why such a simple method would work.

Figure 2 visualizes the intuition of our method: the expected norm of the predicted score for a query image $x^\star$ at time $t$ is effectively the gradient of a Gaussian kernel density estimator (see Appendix A.4). For well-separated points, these estimators' implicit distributions will have peaks at each training point; the gradient vanishes at critical points of the smoothed density, which include the peaks corresponding to the original data points (the blue region in the Fig. 1), up to a bias term from the smoothing. Manipulating this fact allows us to form a simple yet successful estimator.

There should be false positive terms at the other critical points (the green region in Fig 1). In high dimension, these should be by-in-large saddle points between maxima. Because these points occur directly between original datapoints at their arithmetic mean, if the data manifold (the support of $p(x)$) has any curvature, these points would be off manifold; empirically we do not seem to encounter many of them, as indicated by the TPR@1%FPR measurements (see Section 4).

### 3.1 From Forward Diffusion to Score

**Notation:** Let $\{\beta_t\}_{t=1}^{T}$ be the variance schedule of the forward diffusion process in DDPM Ho et al. (2020). Define $\alpha_t := 1 - \beta_t$ and its cumulative product $\bar{\alpha}_t := \prod_{s=1}^{t} \alpha_s$. The total noise variance accumulated up to step $t$ is $\sigma_t^2 := 1 - \bar{\alpha}_t$.

**Forward Diffusion as a Scaled Gaussian Convolution:** With the variance-preserving (VP) schedule of Ho et al. (2020), the forward model is

$$x_t = \sqrt{\bar{\alpha}_t}\, x_0 + \sigma_t \varepsilon, \quad \varepsilon \sim \mathcal{N}(0, I),\ \sigma_t^2 = 1 - \bar{\alpha}_t, \tag{2}$$

where $x_0 \sim p_{\text{data}}$. Marginalizing $x_0$ gives the perturbed data distribution (Song & Ermon, 2019). For a *clean* (without noise) sample $x \in \mathbb{R}^d$,

$$p_t(x) = \int_{\mathbb{R}^d} p_{\text{data}}(x_0)\, \mathcal{N}\big(x \mid \sqrt{\bar{\alpha}_t} x_0, \sigma_t^2 I\big)\, dx_0. \tag{3}$$

Equation 3 is a Gaussian convolution of the original data distribution (see Appendix A.4 for an explicit derivation), followed by a global scaling by $\sqrt{\bar{\alpha}_t}$.

$$p_t(x) = \frac{1}{\bar{\alpha}_t^{d/2}} \left( p_{\text{data}} * \mathcal{N}\left(0, \frac{\sigma_t^2}{\bar{\alpha}_t} I\right)\right)\left(\frac{x}{\sqrt{\bar{\alpha}_t}}\right). \tag{4}$$

Therefore, $p_t(x)$ is the data distribution convolved with scaled Gaussian distribution whose kernel's covariance $\frac{\sigma_t^2}{\bar{\alpha}_t} I = (\bar{\alpha}_t^{-1} - 1)I$ grows monotonically with the timestep $t$.

**Gradient of $p_t(x)$:** Writing the kernel in standard form $\mathcal{K}_t(x, x_0) = (2\pi\sigma_t^2)^{-d/2} \exp(-\|x - \sqrt{\bar{\alpha}_t} x_0\|^2/2\sigma_t^2)$, we can compute its spatial gradient: $\nabla_x \mathcal{K}_t = -\sigma_t^{-2}(x - \sqrt{\bar{\alpha}_t} x_0)\mathcal{K}_t$. We then combine this with Eq. 3 to obtain

$$\nabla_x p_t(x) = -\sigma_t^{-2} \int p_{\text{data}}(x_0)\big(x - \sqrt{\bar{\alpha}_t} x_0\big)\mathcal{K}_t(x, x_0)\, dx_0 \tag{5}$$

which requires continuity assumptions on $p_{\text{data}}$ which are usually assumed by DDPM analyses (Alain & Bengio, 2014).

**Introducing exact likelihood of each datapoint $q_t(x_0 \mid x)$:** Define the exact distribution of the $x_0$ given an observation from the Gaussian smoothed distribution:

$$q_t(x_0 \mid x) = \frac{p_{\text{data}}(x_0)\, \mathcal{K}_t(x, x_0)}{p_t(x)}. \tag{6}$$

Because $p_t(x)$ normalizes Eq. 3, we can then rewrite Eq. 5 as

$$\nabla_x p_t(x) = -\frac{p_t(x)}{\sigma_t^2}\left[ x - \sqrt{\bar{\alpha}_t}\, \underbrace{\mathbb{E}_{q_t(x_0|x)}[x_0|x]}_{\mu_t(x)}\right]. \tag{7}$$

We call $\mathbb{E}_{q_t(x_0|x)}[x_0|x] = \mu_t(x)$ the *denoising mean*; it is the likelihood-weighted average of the positions of the datapoints that could have generated $x$ at time $t$ through the forward process, which is the same as the mean optimal solution to the denoising problem.

**Obtaining the exact score:** Dividing Eq. equation 7 by $p_t(x)$ yields the score of distribution $p_t(x)$ at $x$ (Lemma 6 of Pidstrigach (2022), or, alternatively, applying the chain rule to $\nabla_x \log p_t(x)$ and then substituting in values):

$$s_t(x) = \nabla_x \log p_t(x) = -\frac{x - \sqrt{\bar{\alpha}_t}\,\mu_t(x)}{\sigma_t^2}. \tag{8}$$

This score function $s_t(x)$ is the desired output of the $\varepsilon$-parameterization of Score-based Denoising (Song et al., 2020) and the original DDPM (Ho et al., 2020). During training the UNet is asked to predict the standard noise (Ho et al., 2020):

$$\hat{\varepsilon}_\theta(x, t) \approx \frac{x_t - \sqrt{\bar{\alpha}_t}\,\mu_t(x)}{\sigma_t} = -\sigma_t \nabla_x \log p_t(x) \tag{9}$$

This is consistent with Eq. 151 of Luo (2022). For $x \in \operatorname{supp} p_t$ (with or without noise), the estimator $\hat{\varepsilon}_\theta(x, t)$ should approximate the negative score of $p_t(x)$. However, in practice the data distribution $p_{\text{data}}$ becomes the empirical distribution $p_{\text{training}}$, which is a finite sample of points. The noised distribution $p_t(x)$ is then a kernel smoothing of that empirical distribution, and its finite-sample denoising mean is described in Kamb & Ganguli (2024) and refined in Lukoianov et al. (2025). Equation 3 of Lukoianov et al. (2025) states it as:

$$\mu_t^{\text{finite}}(x) = \sum_{i=1}^{N} w_i(x, t) x_0^{(i)}, \quad w_i(x, t) = \operatorname{softmax}_i \left\{ -\frac{1}{2\sigma_t^2} \|x - \sqrt{\bar{\alpha}} x_0^{(j)}\|_2^2 \right\}_{j \in [N]} \tag{10}$$

where the $x_0^{(i)}$ are training data, and $\operatorname{softmax}_i$ is the $i^{\text{th}}$ index of a softmax function over the $N$ training data points. The discrepancy between the finite sample optimal denoised $x$ distribution and the large sample limite $q_t(x_0|x)$ gives rise to our membership inference attack; the model "overfits" to the training set, and that overfitting gap is the discrepancy which **SimA** seeks to exploit.

## 3.2 MEMBERSHIP INFERENCE ATTACK

Given a datapoint $x \in \mathbb{R}^d$ and a $t = 1, \ldots, T$, our membership decision criterion $\mathcal{A}$ is defined as

$$\mathcal{A}(x, t) = \|\hat{\varepsilon}_\theta(x, t)\|_p. \tag{11}$$

Using $\ell_p$ norms other than $p = 2$ provide slightly improved performance. This trend is also found in another MIA method, Kong et al. (2023). While this is somewhat mysterious, the $\ell_4$ norm appears in sum-of-squares computations (Barak et al., 2015), spherical harmonics (Stanton & Weinstein, 1981), and blind source separation (Hyvarinen, 1997) with surprising regularity.

Following the Bayes-optimal loss–threshold formulation of membership inference in classification models by Sablayrolles et al. (2019), we recast the decision rule for diffusion models. Specifically, we define

$$\mathcal{M}_{\text{opt}}(x, t) = \mathbb{1}\big[\mathcal{A}(x, t) \leq \tau\big], \tag{12}$$

where $\tau$ is a threshold calibrated on a held-out validation set and $\mathcal{A}(x, t) = \|\hat{\varepsilon}_\theta(x, t)\|_2$ is the estimated noise at $x$ for diffusion step $t$. If the predicted noise norm is *smaller* than $\tau$, the sample is inferred to be a *member*; otherwise, it is classified as a *non-member*.

The attack criterion $\mathcal{A}$ can be applied to three cases, which we expand upon below (see appendix A.5 for detailed derivation of the first two cases).

**Case 1 — Member of the Training Set:** Let $x^{(k)}$ denote one of the training images $\{x^{(i)}\}_{i=1}^N$. As $t \to 0$, the finite sample denoising mean collapses to the input (full derivation is in case 1 of Appendix A.5):

$$\mu_t^{\text{finite}}(x^{(k)}) \xrightarrow[t \to 0]{} x^{(k)} \tag{13}$$

Consequently, the estimated noise vector shrinks to zero as well, meaning our criterion $\mathcal{A}(x^{(k)}, t \to 0)$ should be small:

$$\hat{\varepsilon}_\theta(x^{(k)}, t) = \frac{x^{(k)} - \sqrt{\bar{\alpha}_t}\, x^{(k)}}{\sigma_t} \;\sim\; \frac{\sigma_t}{2} x^{(k)} \xrightarrow[t \to 0]{} 0. \tag{14}$$

While the actual value at zero is undefined, and values for small $t < 10$ are empirically unstable for non-member $x^{(k)}$ (leading to a poor estimator, see Appendix A.7 for the detailed reason), we find that for $t \in [10, 300]$, these values will still be smaller than the case 2. These time frames are unfortunately noise schedule and data dependent.

**Case 2 — Held-out but On-Manifold:** Here $x^\dagger$ is sampled from the same data distribution $p_{\text{data}}$ as the training set, yet was *never* shown during training. We use the notation of local moment matching (Bengio et al., 2012; Alain & Bengio, 2014) to describe these points. Consider the local mean defined with respect to that $B_r(x^\dagger)$:

$$\mu_t(x^\dagger)_{\text{local}} = m_r(x^\dagger) = \int_{B_r(x^\dagger)} x p_t(x|x^\dagger) dx \tag{15}$$

where $r \asymp \sigma_t/\sqrt{\bar{\alpha}_t}$. Theorems 2 and 3 of Alain & Bengio (2014) put together produce a statement about a term that is equivalent to $\hat{\varepsilon}$ (Eq. 28 of Alain & Bengio (2014)) (Full derivation is in case 2 of Appendix A.5):

$$m_r(x^\dagger) - x^\dagger \approx \frac{r^2}{d+2} \frac{\partial \log p_t(x)}{\partial x}\bigg|_{x^\dagger} = \frac{r^2}{d+2} \left( \frac{x^\dagger - \sqrt{\bar{\alpha}_t}\mu_t^{\text{finite}}(x^\dagger)}{\sigma_t^2} \right) = -\frac{r^2}{\sigma_t(d+2)} \hat{\varepsilon}_\theta(x^\dagger, t) \tag{16}$$

We claim that for in-support regions of Gaussian mollified empirical distributions with well separated points, these regions will generally not be flat, and thus $\|\hat{\varepsilon}_\theta(x^\dagger, t)\|$ will tend away from zero. The magnitude to which $\|\hat{\varepsilon}_\theta(x^\dagger, t)\|$ diverges from zero clearly depends on the maximal density of the dataset, but as high dimensional spaces have exponentially larger volumes than lower dimensional spaces, even for large datasets (e.g. ImageNet in ResNet resolution) we can expect these voids to be non-trivially large.

**Case 3 — Out-of-Distribution (OOD).** Since no training data support is available in out-of-distribution (OOD) regions, the diffusion model lacks information about these areas. Consequently, the learned score field in such regions is necessarily an extrapolation, and the theoretical derivations established under the in-distribution assumption no longer hold. We do not expect either the diffusion model or our attack criteion to perform well in these regions. First, even though the theoretical finite-sample optimal denoiser is well defined (Eq. 10), a neural network approximation to it will have very little training data in these regions. Second, a trial datapoint $x^*$ is by-definition not from $p_{data}$ in these regions.

**Monte Carlo variant of SimA:** As we mentioned in Figure 2, the intuitive attack method is the expected norm of the predicted score for a query image $x^\star$ at time $t$, i.e. $\mathbb{E}_{\varepsilon \sim \mathcal{N}(0,I)}[\|\hat{\varepsilon}_\theta(x_t, t)\|]$, where $x_t = \sqrt{\bar{\alpha}_t}\, x_0 + \sigma_t \varepsilon$. In fact, it is not hard to see that SimA is a point estimate of this estimator at $\varepsilon = 0$ (mode and median of the standard Gaussian). To evaluate this method, one should resort to Monte Carlo method. Therefore, we name it the **Monte Carlo variant of SimA (SimA-MC).** We originally thought that such a high-dimensional Monte Carlo would perform poorly until a very large number of samples. However, this method turns out to be surprisingly tractable. The Monte Carlo variant of SimA method provides SoTA performance within 30 samples, often within 10. For sufficient ($n \in [10, 30]$) samples, it improves all performance metrics on all datasets of our LDM experiments. The results is summarized in Table 3 and 6. The table results are still incomplete due to rebuttal time for computation. We will make it ready if it goes to camera-ready.

**SimA in comparison to other Diffusion MIA models:** A standard concept in MIA is the use of the training loss function evaluated on the data points in question as the member/non-member

Table 1: Performance of benchmark methods on **DDPM** across four datasets. Best results in **bold**

| Method | #Query↓ | CIFAR-10 (%) | | | CIFAR-100 (%) | | | STL10-U (%) | | | CelebA (%) | | |
|---|---|---|---|---|---|---|---|---|---|---|---|---|---|
| | | ASR↑ | AUC↑ | TPR[†]↑ | ASR↑ | AUC↑ | TPR[†]↑ | ASR↑ | AUC↑ | TPR[†]↑ | ASR↑ | AUC↑ | TPR[†]↑ |
| PIA | 2 | **85.06** | **91.86** | 29.54 | 82.31 | **89.58** | 28.75 | 90.83 | **96.34** | 66.30 | 74.27 | 82.22 | 18.36 |
| PFAMI$_{Met}$ | 20 | 73.64 | 80.32 | 8.16 | 70.58 | 77.05 | 8.44 | 74.78 | 83.74 | 17.00 | 64.34 | 70.18 | 8.13 |
| SecMI$_{stat}$ | 12 | 82.71 | 89.72 | 33.44 | 81.55 | 88.62 | 29.95 | 89.40 | 95.16 | 66.20 | 73.94 | 81.45 | 14.55 |
| Loss | **1** | 77.67 | 84.73 | 24.14 | 74.85 | 82.19 | 19.56 | 84.60 | 91.42 | 47.67 | 69.13 | 75.96 | 15.15 |
| SimA | **1** | 83.62 | 90.45 | **35.86** | **82.51** | 89.85 | **38.84** | **91.15** | **96.34** | **72.75** | **74.90** | **82.85** | **20.86** |

Table 2: Performance of **Guided Diffusion** and **Latent Diffusion** pretrained on ImageNet1K. Best results in **bold**. *Member* set: ImageNet-1K (3000); *Held-out*: ImageNetV2 (3000).

| Method | #Query↓ | ImageNet, Guided | | | ImageNet, LDM | | |
|---|---|---|---|---|---|---|---|
| | | ASR↑ | AUC↑ | TPR[†]↑ | ASR↑ | AUC↑ | TPR[†]↑ |
| PIA | 2 | 64.65 | 66.44 | 9.93 | 55.13 | 56.8 | 2.13 |
| PFAMI$_{Met}$ | 20 | 67.85 | 72.22 | 3.77 | 50.77 | 48.76 | 0.57 |
| SecMI$_{stat}$ | 12 | 77.97 | 82.55 | **34.73** | **56.80** | **56.85** | **2.33** |
| Loss | **1** | 57.27 | 60.38 | 7.00 | 50.48 | 47.9 | 0.6 |
| SimA | **1** | **85.73** | **89.77** | 21.73 | 55.78 | 56.14 | 1.97 |

[†] True Positive Rate at 1 % False Positive Rate.

decision criterion. This is the Loss criterion, and is proposed in Matsumoto et al. (2023). They use a stochastic sample $\varepsilon$ to estimate this criterion, adding it to the test point $x^*$.

$$\text{Loss} = \|\varepsilon - \hat{\varepsilon}_\theta(\sqrt{\bar{\alpha}_t}x^* + \sqrt{1 - \bar{\alpha}_t}\varepsilon, t)\| \quad \text{(Matsumoto et al., 2023)} \tag{17}$$

In theory this should be evaluated across a large number of $\varepsilon$ measurements, but for each point they choose to use only one. This method is predicated on the idea that for points in the training set, the noise estimation will be better or even overfit in comparison to points not in the training set. **SimA** is the evaluation of this method at $\varepsilon = 0$, which is the mean and mode of the $\varepsilon$ distribution. Effectively, Matsumoto et al. (2023) are measuring draws from lower likelihood areas, which may not exhibit the overfit phenomenon as well as $\varepsilon = 0$. Fu et al. (2023) provide this same loss estimate as the selection criterion, but increase the Monte Carlo sampling to 20 and perform it only for a single step, sampling $\varepsilon_t$ the stepwise noise instead of sampling $\varepsilon$.

SecMI of Duan et al. (2023) takes this one step further, evaluating not only a score term but also a single step term, which measures sensitivity to single step differences in $t$:

$$\text{SecMI} = \|\sqrt{1 - \bar{\alpha}}(\hat{\varepsilon}_\theta(x, t) - \hat{\varepsilon}_\theta(\sqrt{\bar{\alpha}_{t+1}}x^* + \sqrt{1 - \bar{\alpha}_{t+1}}\varepsilon, t + 1))\| \tag{18}$$

SecMI is dependent on sampling $\varepsilon$'s as well; the authors prescribe using $N = 12$ samples.

The closely related PIA method computes this same loss quantity again, but using a $t = 0$ term instead of the $\varepsilon$ sample.

$$\text{PIA} = \|\hat{\varepsilon}_\theta(x, t = 0) - \hat{\varepsilon}_\theta(\sqrt{\bar{\alpha}_t}x^* + \sqrt{1 - \bar{\alpha}_t}\hat{\varepsilon}_\theta(x, t = 0), t)\| \tag{19}$$

While in theory diffusion models might not be well defined at $t = 0$, in practice they often can extrapolate as they are trained on nearby $t$; in the continuous time case they are trained on the interval $[0, 1]$. Our method has similar components to this method, manipulating $t$ around the test point, but again replacing its "ground-truth" noise with $\varepsilon = 0$ (here replaced by $\hat{\varepsilon}(x_0, t = 0)$).

## 4 EXPERIMENTS

### 4.1 SETUP

We evaluated our attack on **15 member–held-out pairs** drawn from **11 datasets** (see Appendix A.1). The experiments were conducted on the following target models:

Table 3: Attack performance of four baseline methods used to evaluate memorization of **LDMs**. Apart from SimA(#mc), best results of in **bold**. The gray area denotes the Monte Carlo variant of SimA. #mc denotes the number of Monte Carlo samples. At #mc=10, SimA advances the state-of-the-art across all reported metrics. For ImageNet, ImageNet-1K (train split); *Held-out*: ImageNetV2 (validation split).

| Method | #Query↓ | CIFAR-10 (%) | | | CelebA (%) | | | ImageNet (%) | | |
|---|---|---|---|---|---|---|---|---|---|---|
| | | AUC↑ | ASR↑ | TPR@1%FPR↑ | AUC↑ | ASR↑ | TPR@1%FPR↑ | AUC↑ | ASR↑ | TPR@1%FPR↑ |
| Loss | 1 | 73.28 | 67.48 | 7.86 | 68.65 | 63.47 | 6.39 | 67.49 | 63.16 | **4.09** |
| SecMI | 12 | 87.42 | 80.12 | 19.11 | 83.09 | 75.85 | 10.53 | 68.21 | 63.44 | 3.71 |
| PIA | 2 | 85.73 | 78.19 | 15.90 | 83.36 | 75.87 | 9.41 | 66.09 | 62.52 | 2.95 |
| SimA ($\varepsilon = 0$) | 1 | **89.10** | **81.63** | **19.88** | **84.66** | **77.04** | **11.09** | 69.62 | 64.92 | 3.87 |
| SimA (#mc=10) | 10 | **90.68** | **82.71** | **39.86** | **91.55** | **83.56** | **39.76** | **71.13** | **65.95** | **7.44** |

**Denoising Diffusion Probabilistic Model:** For CIFAR-10, CIFAR-100 (Krizhevsky et al., 2009), STL10-U (unlabeled split) (Coates et al., 2011), and CelebA (Liu et al., 2015), we trained a vanilla DDPM (Ho et al., 2020) from scratch on the *member* set. From each training split we subsample $n$ images and partition them equally into a *member* set and a *held-out* set.

**Pre-trained Guided Diffusion:** We examined the publicly released Guided Diffusion model[1] (Dhariwal & Nichol, 2021) trained on ImageNet-1K (Russakovsky et al., 2015). ImageNetV2 (Recht et al., 2019), collected to mirror the original distribution (same data collection process and same year range), serves as the *held-out* set.

Full statistics including dataset splits, resolutions, etc. are summarized in Table 4 of Appendix A.1. Additional experiments on LDM models (Stable Diffusion) similar to the experiments in Zhai et al. (2024) are included in Appendix A.3. These exhibit the same low performance across all methods as the LDM on ImageNet experiments.

**Baselines:** Earlier membership-inference attacks (MIAs) aimed at GANs and VAEs—e.g., (Chen et al., 2020; Hilprecht et al., 2019; Hu & Pang, 2021)—do not transfer well to diffusion models, as shown by Duan et al. (2023). Consequently, we restricted our evaluation to attacks specifically designed for diffusion models. We compared our method with four baselines: SecMI (Duan et al., 2023), PIA (Kong et al., 2023), PFAMI (Fu et al., 2023), Loss (Matsumoto et al., 2023). We omitted CLiD (Zhai et al., 2024), a text-conditioned MIA, because its text-supervision is incompatible with the setup used in most of our experiments, and the provided code is not usable.

**Evaulation Metrics:** We evaluated attack performance using several metrics: ASR (attack success rate, i.e., membership inference accuracy), AUC (Area Under ROC Curve), TPR at 1% FPR (TPR@1%FPR), and the number of queries per attack (#Query). The TPR@1%FPR is computed by selecting the threshold $\tau$ at which the false positive rate falls just below $0.01$, and reporting the corresponding true positive rate at that operating point. The ASR is defined as the maximum accuracy achieved over all thresholds, i.e. $\tau^* = \max_\tau \frac{1}{2}\big(\text{TPR}(\tau) + 1 - \text{FPR}(\tau)\big)$ in the balanced setting. The AUC is computed as the trapezoidal integral of $\text{TPR}(\tau)$ against $\text{FPR}(\tau)$ across all thresholds.

**Implementation details:** Some baselines (Duan et al., 2023; Kong et al., 2023; Zhai et al., 2024) augmented their score- or feature-vector statistic with an auxiliary neural classifier; to focus on the statistic itself we evaluated only the norm-based versions, which were SecMI$_\text{stat}$ and PFAMI$_\text{Met}$. Of the two variants (PIA and PIAN) introduced by Kong et al. (2023), we benchmarked only PIA, as PIAN showed no statistically significant gain in general from their experiments.

To minimise re-implementation error, our codebase reused the official releases of SecMI, PIA, and Guided Diffusion wherever possible. We failed to re-use PFAMI's code as the provided code was inoperable. A reimplemented copy is provided in our code base. PFAMI, in several cases, failed to attack the victim model (ASR $\approx 50\%$). We hypothesize that this degradation arose from the sensitivity of its Monte Carlo estimator to the effective sample size, which varied with dataset characteristics and latent dimensionality, yielding high-variance estimates. For each method, we followed the hyperparameter suggestion in their original paper. Notably, $l_2$-norm, $l_4$-norm and $l_2$-norm were used for SecMI, PIA and Loss as suggested. $l_4$-norm was used for SimA as it achieved the best performance in general. For the timestep-dependent attacks (SimA, SecMI, PIA, Loss), we swept

---

[1]https://github.com/openai/guided-diffusion

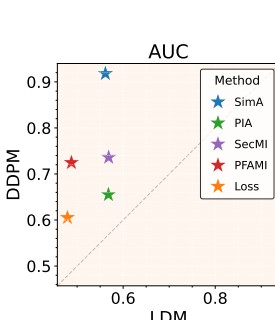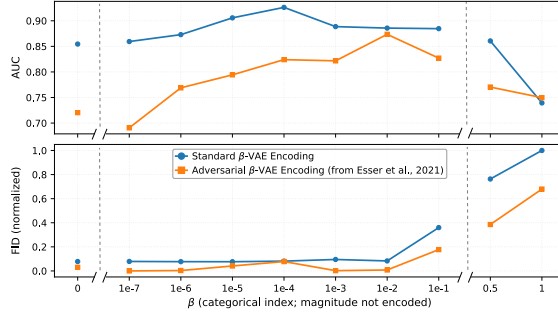

Figure 3: **Left:** The performance comparison of AUC between DDPM and Latent Diffusion Model on the same *member/held-out* splits for ImageNet-1K and ImageNetV2. **Right:** Comparison of AUC of MIA (top) and normalized FID (bottom) across categorical $\beta$ settings for Standard $\beta$-VAE (Burgess et al., 2018) Encoding and Adversarial $\beta$-VAE (Esser et al., 2021) on CIFAR-10. The x-axis represents categorical indices of $\beta$ values, without encoding their numerical magnitude.

$t = 0:300$ and reported the best-performing value for each method (PFAMI is timestep-free and therefore needed no sweep). More implementation details can be found in Section 4.2.

## 4.2 MAIN RESULTS

**In Table 1**, all DDPM baselines were trained on a member split and evaluated on an held-out split of equal sizes: $25\,\mathrm{k}/25\,\mathrm{k}$ for CIFAR-10 and CIFAR-100, $50\,\mathrm{k}/50\,\mathrm{k}$ for STL10-U, and $30\,\mathrm{k}/30\,\mathrm{k}$ for CelebA, all at a spatial resolution of $32 \times 32$. We adopted the public checkpoints and splits released by SecMI for CIFAR-10/100, and retrained STL10-U and CelebA for $18\,\mathrm{k}$ and $60\,\mathrm{k}$ steps, respectively. Across all datasets and metrics, SimA achieved the best in almost all the experiments over three metrics (ASR, AUC, TPR@1%FPR) while requiring the fewest queries, underscoring its efficiency and practical advantage.

**In Table 2**, we further evaluated Guided Diffusion (Dhariwal & Nichol, 2021) and LDM (Rombach et al., 2022) at their public class-conditional ImageNet-1K checkpoints ($256 \times 256$). We find that for Guided Diffusion **SimA** has the highest ASR and AUC by a wide margin, while SecMI has a very high TPR@1%FPR. For the Latent Diffusion *on the same dataset and holdout*, SecMI technically has the best performance, but all methods are extremely bad, achieving less than 57 AUC and 2.3% TPR@1%FPR.

**DDPM *vs.* Latent Diffusion Model:** In the original LDM (Rombach et al., 2022), the encoder is a VQ-VAE (Van Den Oord et al., 2017); its discrete code-book forces quantization on patches. A continuous VAE (Kingma et al., 2013) is used in Stable Diffusion and passes richer detail, but still compresses pixel-level information. In both cases, embeddings may express high-level coarse-grained information, and pathological memorization of pixel patterns might be reduced. To investigate this, recap from experiments on Table 2, we used the same *member* and *held-out* sets on Guided Diffusion Model and LDM, and All other variables (seed, code, versions) are held constant.

As shown in Fig. 3 (Left) and Table 2, performance drops markedly from DDPM to LDM: SimA, PIA, and SecMI$_{\text{stat}}$ fall to $\sim$55% in ASR, 56% in AUC, and 2% in TPR@1%FPR, while PFAMI$_{\text{Met}}$ and Loss perform no better than random guessing.

To further investigate why this particular LDM checkpoint exhibits reduced vulnerability to MIA, we conduct a controlled study on CIFAR-10. We train a series of VAEs with varying $\beta$ values in the $\beta$-VAE framework (Burgess et al., 2018). The LDMs are subsequently trained based on these frozen VAE encoders. We perform attack on latent diffusion model. As shown in Fig. 3 (Right-Top), MIA performance is sensitive to the KL-divergence weight $\beta$. Introducing a small KL regularization on VAE training increases AUC, but stronger constraints lead to a monotonic decline, consistent with the expectation that a tighter information bottleneck reduces membership leakage during encoding. Importantly, however, the FID performance is relatively unaffected by the $\beta$ parameter until much

larger values, indicating that it is possible to preserve generative quality (at least up to the FID metric) while also increasing robustness to MIA.

Interestingly, incorporating a discriminator between the input image and reconstructed image during VAE training, an approach originally developed for VQ-GAN (Esser et al., 2021), also reduces MIA effectiveness in LDMs while improving sample fidelity. As illustrated in Fig. 3 (Right), this minor architectural modification yields models that are both more robust to membership inference and capable of generating images with lower FID relative to the data distribution. Full training details are provided in Section A.6 of the Appendix.

This series of experiments, as well as the disparate results on ImageNet-1K between image-domain diffusion and a latent domain diffusion model indicate a gap in the model inversion literature. Previously models have focused on the Diffusion process itself; our results in Table 2 indicate that Diffusion processes on their own can already be solved for MIA to a high degree of fidelity with a very simple estimator. On the otherhand, the LDM class of models have MIA performance only marginally above at-random. We believe that this should be the focus of model inversion efforts moving forward.

## 5 CONCLUSION

In the present work we have described a simple membership inference estimator, giving theoretical justification for its performance, and for the performance of similar estimators in the literature which previously lacked a unified theoretical backing. We demonstrate that this estimator has competitive performance on many baselines, including state-of-the-art performance on ImageNet-1K image-domain inversion. Our experiments also elucidate a hole in the current literature concerning latent-domain diffusion methods, where all tested membership inference methods currently fail.

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

## A    Appendix

### A.1    Datasets and splits

Datasets and splits used for our experiments are summarized in Table 4.

### A.2    Performance rank across 15 experiments

A rank distribution over 5 benchmark methods (Duan et al., 2023; Kong et al., 2023; Matsumoto et al., 2023; Fu et al., 2023) across 15 experiments is provided in Figure 5.

### A.3    Additional experiments on Class Conditional ImageNet, and Stable Diffusion pre-trained on LAION-Aesthetics v2 5+.

For the unconditional case of Guided Diffusion, please see Table 5.

**Pre-trained Latent Diffusion Models:** For Pokémon[2], COCO2017-Val (Lin et al., 2014), and Flickr30k (Young et al., 2014), we fine-tuned Stable Diffusion v1-4[3] on a randomly selected subset of each training split, reserving an equally sized subset as the held-out set. We also studied the original Stable Diffusion v1-5[4] checkpoint, pre-trained on LAION-Aesthetics v2 5+ (Schuhmann et al., 2022) (*member* set). Here we sampled 2500 images from LAION-2B-MultiTranslated[5] and COCO2017-Val as non-members, respectively. Notably, the images from LAION-2B-MultiTranslated are filtered with attributes *pwatermark* $< 0.5$; *prediction (aesthetic_score)* $> 5.0$ and *similarity* $> 0.3$. *pwatermark* and *prediction* are to minimize the domain shift between the member set and the held-out set. And *similarity* is to ensure the alignment of the text-image pairs.

Stable Diffusion v1-4 was fine-tuned for **15k**, **50k**, and **200k** steps on the member splits of Pokémon, COCO-2017-Val, and Flickr30k, respectively, with a fixed learning rate of $1 \times 10^{-5}$ (AdamW). None of these datasets are in the original pre-training corpus. Additionally, we adopted the default data augmentation (Random-Crop and Random-Flip) while training.

The results are summarized on Table 7 and 8, which followed the experimental protocol of Zhai et al. (2024) and evaluated membership-inference attacks on Stable Diffusion under two scenarios:

*Fine-tuning.* A Stable Diffusion v1-4 checkpoint was fine-tuned on the designated *member* split of each target dataset; attacks were launched on paired *member/held-out* splits.

*Pre-training.* A pre-trained Stable Diffusion v1-5 model was attacked directly, without additional fine-tuning. The *member* set was a subset of LAION-Aesthetics v2 5+ that was used during pre-training, while the *held-out* set was an equally-sized split drawn from the target dataset.

For every dataset we created five random *member/held-out* partitions. All experiments were run in both text-conditional and unconditional modes, except on the Pokémon dataset where only the conditional mode was considered because the model over-fit rapidly. The unconditional baseline was obtained by passing an empty string to the CLIP text encoder. For datasets evaluated in both modes, the absolute gain of the conditional attack over the unconditional one was reported in blue. Across all datasets and metrics, conditioning on text consistently strengthened the attack, indicating that text-conditional generation memorised training data more severely than unconditional generation.

---

[2] https://huggingface.co/datasets/lambdalabs/pokemon-blip-captions
[3] https://huggingface.co/CompVis/stable-diffusion-v1-4
[4] https://huggingface.co/stable-diffusion-v1-5/stable-diffusion-v1-5
[5] https://huggingface.co/datasets/laion/laion2B-multi-joined-translated-to-en

Table 4: Datasets and splits used for our experiments.

| Model | Member | Held-out | Pre-trained | Fine-tuned | Splits | Resolution | Cond. |
|---|---|---|---|---|---|---|---|
| DDPM | CIFAR-10 | CIFAR-10 | No | – | 25k/25k | 32 | – |
| | CIFAR-100 | CIFAR-100 | No | – | 25k/25k | 32 | – |
| | STL10-U | STL10-U | No | – | 50k/50k | 32 | – |
| | CelebA | CelebA | No | – | 30k/30k | 32 | – |
| Guided Diffusion | ImageNet-1k | ImageNetV2 | Yes | No | 3k/3k | 256 | class |
| Stable Diffusion V1-4 | Pokémon | Pokémon | Yes | Yes | 416/417 | 512 | text |
| | COCO2017-Val | COCO2017-Val | Yes | Yes | 2.5k/2.5k | 512 | text |
| | Flickr30k | Flickr30k | Yes | Yes | 10k/10k | 512 | text |
| Stable Diffusion V1-5 | LAION-Aesthetics v2 5+ | LAION-2B-MultiTranslated | Yes | No | 2.5k/2.5k | 512 | text |
| | LAION-Aesthetics v2 5+ | COCO2017-Val | Yes | No | 2.5k/2.5k | 512 | text |

## A.4 FROM MARGINAL DENSITY TO EXPLICIT GAUSSIAN CONVOLUTION

Let $\bar{\alpha}_t = \prod_{s=1}^{t} \alpha_s$ and $\sigma_t^2 = 1 - \bar{\alpha}_t$. The VP forward marginal is

$$p_t(x) = \int_{\mathbb{R}^d} p_{\text{data}}(x_0) \, \mathcal{N}\big(x \mid \sqrt{\bar{\alpha}_t} \, x_0, \ \sigma_t^2 I\big) \, dx_0. \tag{20}$$

Set $u = \sqrt{\bar{\alpha}_t} \, x_0$ so that $x_0 = u/\sqrt{\bar{\alpha}_t}$ and $dx_0 = \bar{\alpha}_t^{-d/2} \, du$. Then

$$p_t(x) = \bar{\alpha}_t^{-d/2} \int_{\mathbb{R}^d} p_{\text{data}}\left(\frac{u}{\sqrt{\bar{\alpha}_t}}\right) \, \mathcal{N}\big(x \mid u, \ \sigma_t^2 I\big) \, du. \tag{21}$$

Introduce $\tilde{x} := x/\sqrt{\bar{\alpha}_t}$ and $\tilde{u} := u/\sqrt{\bar{\alpha}_t}$, so $u = \sqrt{\bar{\alpha}_t} \, \tilde{u}$ and $du = \bar{\alpha}_t^{d/2} \, d\tilde{u}$. Substituting into equation 21 cancels the Jacobians and yields

$$p_t(x) = \int_{\mathbb{R}^d} p_{\text{data}}(\tilde{u}) \, \mathcal{N}\big(\sqrt{\bar{\alpha}_t} \, (\tilde{x} - \tilde{u}) \mid 0, \ \sigma_t^2 I\big) \, d\tilde{u}. \tag{22}$$

Use the Gaussian scaling identity

$$\mathcal{N}(az \mid 0, \ \sigma^2 I) = a^{-d} \mathcal{N}\left(z \mid 0, \ \frac{\sigma^2}{a^2} I\right) \qquad (\text{for } a > 0),$$

with $a = \sqrt{\bar{\alpha}_t}$ and $z = \tilde{x} - \tilde{u}$. Then equation 22 becomes

$$p_t(x) = \bar{\alpha}_t^{-d/2} \int_{\mathbb{R}^d} p_{\text{data}}(\tilde{u}) \, \mathcal{N}\left(\tilde{x} - \tilde{u} \mid 0, \ \frac{\sigma_t^2}{\bar{\alpha}_t} I\right) \, d\tilde{u}. \tag{23}$$

The integral in equation 23 is a (Euclidean) convolution evaluated at $\tilde{x}$:

$$\big(p_{\text{data}} * \mathcal{N}(0, \tfrac{\sigma_t^2}{\bar{\alpha}_t} I)\big)(\tilde{x}) = \int_{\mathbb{R}^d} p_{\text{data}}(\tilde{u}) \, \mathcal{N}\left(\tilde{x} - \tilde{u} \mid 0, \ \tfrac{\sigma_t^2}{\bar{\alpha}_t} I\right) \, d\tilde{u}.$$

Therefore,

$$p_t(x) = \bar{\alpha}_t^{-d/2} \left(p_{\text{data}} * \mathcal{N}\big(0, \tfrac{\sigma_t^2}{\bar{\alpha}_t} I\big)\right)\left(\frac{x}{\sqrt{\bar{\alpha}_t}}\right) \tag{24}$$

with $\sigma_t^2/\bar{\alpha}_t = \bar{\alpha}_t^{-1} - 1$.

## A.5 MIA IN THREE CASES

When the *data prior* is the empirical distribution constructed from the training set $\{x^{(i)}\}_{i=1}^{N}$. We have $p_{\text{train}}(x_0) = \frac{1}{N} \sum_i \delta(x_0 - x^{(i)})$. For any test image $x$, the UNet predicts

$$\hat{\varepsilon}_\theta(x, t) \approx \frac{x - \sqrt{\bar{\alpha}_t} \, \mu_t(x)}{\sigma_t}, \tag{25}$$

Table 5: Performance of **Guided Diffusion** pretrained on ImageNet1K with class conditional changes. Best results in **bold**, second-best underlined. *Member* set: ImageNet-1K (3000); *Held-out*: ImageNetV2 (3000). The performance difference of the class-conditional attack from the unconditional one is in blue. Format: *uncond.% ($\Delta = cond.-uncond.$)*.

| Method | #Query↓ | ASR↑ | AUC↑ | TPR@1%FPR↑ |
|---|---|---|---|---|
| PIA | 2 | 64.65(-1.98) | 66.44(-0.93) | 9.93(-3.96) |
| PFAMI$_{\text{Met}}$ | 20 | 67.85(+0.22) | 72.22(+0.25) | 3.77(+0.50) |
| SecMI$_{\text{stat}}$ | 12 | 77.97(-9.09) | 82.55(-8.94) | **34.73**(-19.53) |
| Loss | 1 | 57.27(+0.10) | 60.38(+0.15) | 7.00(+0.07) |
| SimA | 1 | **85.73**(+2.57) | **89.77**(+2.03) | 21.73(+6.87) |

Table 6: Updated table, we leave the old table for rebuttal use. See our rebuttal for additional explanation.This table is still incomplete; we will make it ready if it goes to camera-ready. Attack performance of four baseline methods used to evaluate memorization of **Stable Diffusion**. Apart from SimA(#mc), best results in **bold**. The gray area is the Monte Carlo variant of SimA. #mc denotes the number of Monte Carlo samples. Notably, at #mc=10 and 30, SimA pushes the state-of-the-art forward across all reported metrics.

| Method | #Query↓ | Pokémon (%) | | | MS-COCO (%) | | | Flickr (%) | | |
|---|---|---|---|---|---|---|---|---|---|---|
| | | AUC↑ | ASR↑ | TPR@1%FPR↑ | AUC↑ | ASR↑ | TPR@1%FPR↑ | AUC↑ | ASR↑ | TPR@1%FPR↑ |
| Loss | 1 | 92.03 | 85.47 | **40.77** | 83.26 | 75.80 | 13.48 | 63.13 | 59.74 | 2.35 |
| SecMI | 12 | 88.12 | 81.04 | 29.02 | 91.35 | 84.00 | **39.16** | **71.81** | **66.54** | **5.29** |
| PIA | 2 | **94.93** | **90.52** | 32.85 | 92.51 | 85.72 | 24.76 | 68.88 | 64.61 | 2.67 |
| SimA ($\varepsilon = 0$) | 1 | 93.50 | 87.87 | 20.38 | **93.71** | **87.34** | 29.80 | 70.04 | 65.96 | 2.59 |
| SimA (#mc=10) | 10 | **96.75** | **91.47** | **63.79** | 93.21 | 85.76 | 45.36 | 70.26 | 65.65 | 4.47 |
| SimA (#mc=30) | 30 | **97.01** | **92.31** | **70.50** | 94.24 | 86.66 | 52.48 | 72.23 | 66.85 | 6.33 |

We are interested in the quantity

$$\mu_t^{\text{finite}}(x) = \mathbb{E}_{q_t(x_0|x)}[x_0] = \sum_{i=1}^{N} w_i(x,t)\, x^{(i)},$$

$$w_i(x,t) = \frac{\exp[-\|x - \sqrt{\bar{\alpha}_t}\, x^{(i)}\|^2/(2\sigma_t^2)]}{\sum_{j=1}^{N} \exp[-\|x - \sqrt{\bar{\alpha}_t}\, x^{(j)}\|^2/(2\sigma_t^2)]}. \tag{26}$$

Equations equation 26 express the posterior mean as a weighted average of the training samples, where each kernel weight $w_i(x,t)$ depends on the Euclidean distance between the noisy query $x$ and the down-scaled datum $\sqrt{\bar{\alpha}_t}\, x^{(i)}$.

CASE 1 — TRAINING MEMBER

Pick $x = x^{(k)} \in \{x^{(i)}\}$. Set $x = x^{(k)}$ in equation 26. Define the squared distances

$$d_{ik}(t) := \left\|x^{(k)} - \sqrt{\bar{\alpha}_t}\, x^{(i)}\right\|^2, \qquad \Delta_{ik}(t) := \frac{d_{ik}(t) - d_{kk}(t)}{2\sigma_t^2}. \tag{27}$$

Using equation 26 and equation 27 we obtain

$$w_k(x^{(k)},t) = \left[1 + \sum_{i \neq k} \exp(-\Delta_{ik}(t))\right]^{-1}, \tag{28}$$

$$w_{i \neq k}(x^{(k)},t) = \exp(-\Delta_{ik}(t))\, w_k(x^{(k)},t). \tag{29}$$

SMALL-NOISE LIMIT $t \to 0$. Because $\sigma_t^2 = 1 - \bar{\alpha}_t \to 0$ and

$$1 - \sqrt{\bar{\alpha}_t} = O(\sigma_t^2), \qquad d_{kk}(t) = \left(1 - \sqrt{\bar{\alpha}_t}\right)^2 \|x^{(k)}\|^2 = O(\sigma_t^4),$$

Table 7: Performance of **Stable Diffusion** that is pretrained on LAION-Aesthetics v2 5+. COCO2017-Val and Flickr30k here follows the fine-tuning setting. LAION-2B-MultiTranslated is the pre-training setting. The performance difference of the text-conditional attack from the unconditional one is in blue. Format: *uncond.%* ($\Delta = cond.-uncond.$).

| Method | #Query↓ | COCO2017-Val (%) | | | Flickr30k (%) | | | LAION-2B-MultiTranslated (%) | | |
|---|---|---|---|---|---|---|---|---|---|---|
| | | ASR↑ | AUC↑ | TPR$^\dagger$↑ | ASR↑ | AUC↑ | TPR$^\dagger$↑ | ASR↑ | AUC↑ | TPR$^\dagger$↑ |
| PIA | 2 | 62.4(+0.7) | 65.8(+1.2) | 2.88(+0.32) | 59.3(+0.6) | 61.8(+0.8) | 1.64(+0.17) | **60.6**(–0.0) | **63.8**(+0.0) | **4.12**(+0.08) |
| PFAMI$_{\text{Met}}$ | 20 | 61.9(+0.7) | 65.4(+0.9) | 2.28(+0.20) | 59.2(+0.4) | 61.6(+0.6) | 1.75(+0.11) | 50.0(+0.0) | 40.8(+0.2) | 0.80(+0.08) |
| SecMI$_{\text{stat}}$ | 12 | **64.4**(+0.7) | **69.1**(+1.0) | **4.32**(+0.60) | **61.1**(+0.5) | **64.6**(+0.9) | 2.85(+0.57) | 56.7(+0.1) | 58.6(+0.0) | 2.92(-0.16) |
| Loss | **1** | 58.5(+0.6) | 61.7(+1.0) | 3.44(+0.12) | 56.1(+0.4) | 58.3(+0.6) | **1.83**(+0.08) | 50.1(+0.0) | 37.0(+0.0) | 0.80(+0.00) |
| SimA | **1** | 63.1(+0.9) | 66.5(+1.3) | 3.04(+0.28) | 60.1(+0.6) | 62.5(+0.9) | 1.71(+0.17) | 60.3(-0.0) | 63.3(-0.0) | 3.20(-0.08) |

$^\dagger$ True Positive Rate at 1 % False Positive Rate

Table 8: Performance of **Stable Diffusion** pre-trained on LAION-Aesthetics v2 5+. Pokémon uses SD v1-4 (fine-tuned) and reports *conditional only* (no parentheses). COCO2017-Val uses SD v1-5 (held-out) and reports in the format *uncond.%* ($\Delta = cond. - uncond.$). Best in **bold**, second-best underlined.

| Method | #Query↓ | Pokémon (fine-tune) (%) | | | COCO2017-Val (pre-training) (%) | | |
|---|---|---|---|---|---|---|---|
| | | ASR↑ | AUC↑ | TPR$^\dagger$↑ | ASR↑ | AUC↑ | TPR$^\dagger$↑ |
| PIA | 2 | **89.9** | **94.6** | 30.9 | 53.7(-0.06) | 52.3(-0.14) | **2.1**(-0.12) |
| PFAMI$_{\text{Met}}$ | 20 | 50.0 | 19.0 | 0.0 | 52.0(+0.08) | 48.7(+0.12) | 0.4(+0.00) |
| SecMI$_{\text{stat}}$ | 12 | 81.4 | 87.8 | **34.1** | **55.7**(-0.06) | **56.0**(-0.13) | 1.2(+0.08) |
| Loss | 1 | 80.2 | 87.9 | 24.7 | 55.4(+0.02) | 49.3(-0.01) | 0.4(+0.00) |
| SimA | 1 | 87.6 | 93.0 | 21.8 | 54.5(-0.04) | 53.7(-0.07) | 1.5(+0.12) |

$^\dagger$ True Positive Rate at 1 % False Positive Rate.

we have, for $i \neq k$,

$$\Delta_{ik}(t) \sim \frac{\|x^{(k)} - x^{(i)}\|^2}{2\sigma_t^2} \xrightarrow[t \to 0]{} +\infty, \qquad \Delta_{kk}(t) = 0.$$

Hence

$$w_k(x^{(k)}, t) \xrightarrow[t \to 0]{} 1, \qquad w_{i \neq k}(x^{(k)}, t) \xrightarrow[t \to 0]{} 0. \tag{30}$$

This implies

$$\boxed{\mu_t^{\text{finite}}(x^{(k)}) \xrightarrow[t \to 0]{} x^{(k)}}. \tag{31}$$

Moreover, substituting into the estimator equation 25,

$$\left\|\hat{\varepsilon}_\theta(x^{(k)}, t)\right\|_2 \approx \left\|\frac{x^{(k)} - \sqrt{\bar{\alpha}_t}\, x^{(k)}}{\sigma_t}\right\|_2 = \frac{|1 - \sqrt{\bar{\alpha}_t}|}{\sigma_t} \|x^{(k)}\|_2. \tag{32}$$

To see the asymptotic form using Taylor expansion around $\sigma_t^2 = 0$, note that

$$\sqrt{\bar{\alpha}_t} = \sqrt{1 - \sigma_t^2} = 1 - \tfrac{1}{2}\sigma_t^2 - \tfrac{1}{8}\sigma_t^4 + O(\sigma_t^6),$$

so that

$$1 - \sqrt{\bar{\alpha}_t} = \tfrac{1}{2}\sigma_t^2 + O(\sigma_t^4).$$

Therefore

$$\frac{1 - \sqrt{\bar{\alpha}_t}}{\sigma_t} = \tfrac{1}{2}\sigma_t + O(\sigma_t^3),$$

and hence

$$\left\|\hat{\varepsilon}_\theta(x^{(k)}, t)\right\|_2 \sim \tfrac{\sigma_t}{2} \|x^{(k)}\|_2 \xrightarrow[t \to 0]{} 0. \tag{33}$$

CASE 2 — HELD-OUT BUT ON-MANIFOLD

Consider a test data point $x^\dagger$ which is not in our original dataset (i.e., $x^\dagger \notin \{x^{(i)}\}_{i=1}^N$), but sampled from the same generating distribution $p_{\text{data}}$. Under the diffusion process assumptions, the local weighted mean $\mu_t(x)$ in equation 26 has the same algebraic form as a kernel regression (Nadaraya, 1964; Watson, 1964), using training dataset $\{x^{(i)}\}_{i=1}^N$ with (effective) Gaussian bandwidth

$$h(t) := \frac{\sigma_t}{\sqrt{\bar{\alpha}_t}} . \tag{34}$$

The weights $w_i(x, t)$ in equation 26 are proportional to $\exp\left(-\|x^\dagger - \sqrt{\bar{\alpha}}x^{(i)}\|_2^2/(2h(t)^2)\right)$, so $\mu_t^{\text{finite}}(x^\dagger)$ coincides with a Gaussian–kernel local average of "nearby" training points, where "nearby" is on the order of bandwidth $h(t)$. The kernel-weighted local mean with radius $r$ around $x^\dagger$ is defined as

$$m_r(x^\dagger) := \frac{\displaystyle\int_{B_r(x^\dagger)} u\, K_r(u - x^\dagger)\, p(u)\, du}{\displaystyle\int_{B_r(x^\dagger)} K_r(u - x^\dagger)\, p(u)\, du}, \tag{35}$$

$$K_r(z) \propto \exp\left(-\frac{\|z\|^2}{2r^2}\right), \tag{36}$$

where $r \asymp h(t)$ and $p(u)$ is the empirical distribution. This is a normalization of the local mean defined in Eq. 15 of this paper, and Appendix 6.4.1 of Alain & Bengio (2014). By local moment matching (Bengio et al., 2012; Alain & Bengio, 2014), combining Theorems 2 and 3 of Alain & Bengio (2014) (see their Eq. (28)) yields the second-order expansion

$$m_r(x^\dagger) - x^\dagger = \frac{r^2}{d+2} \left.\frac{\partial \log p_t(x)}{\partial x}\right|_{x^\dagger} + o(r^3), \tag{37}$$

where $p_t$ is the Gaussian-mollified density at scale $\sigma_t$. Using equation 9, we obtain the proportionality stated in the main text:

$$\boxed{m_r(x^\dagger) - x^\dagger \approx \frac{r^2}{d+2} \left.\frac{\partial \log p_t(x)}{\partial x}\right|_{x^\dagger} = \frac{r^2}{d+2} \left(\frac{x^\dagger - \sqrt{\bar{\alpha}_t}\mu_t^{\text{finite}}(x^\dagger)}{\sigma_t^2}\right) = -\frac{r^2}{\sigma_t(d+2)}\hat{\varepsilon}_\theta(x^\dagger, t)} \tag{38}$$

Notably, this equation should *not* be correct if $x^\dagger$ is in a low density region of the support of $p_{data}$, but these points should generally be rarely sampled by definition. Moreover, the $\hat{\varepsilon}$ estimate of the score will also be extrapolating at those points.

**Implication for the attack statistic.** In in-support regions of the empirical distribution, the local geometry is generally *not* flat, so $\|m_r(x^\dagger) - x^\dagger\| > 0$ for $r > 0$ sufficiently small. By equation 38, this implies $\|\hat{\varepsilon}_\theta(x^\dagger, t)\|$ is bounded away from zero (at fixed $t$), hence $\mathcal{A}(x^\dagger, t) = \|\hat{\varepsilon}_\theta(x^\dagger, t)\|_p$ exceeds the member case (Case 1). This matches the intuition summarized in the main text: on-manifold, held-out queries denoise less precisely than memorized training points, yielding a moderately larger attack statistic without the divergence seen off-manifold.

CASE 3 — OUT-OF-DISTRIBUTION (FAR OFF-MANIFOLD)

Since no training data support is available in out-of-distribution (OOD) regions, the diffusion model lacks information about these areas. Consequently, the learned score field in such regions is necessarily an extrapolation, and the theoretical derivations established under the in-distribution assumption no longer hold.

## A.6 TRAINING DETAILS OF ADVERSARIAL $\beta$-VAE ON CIFAR-10

We train a series of $\beta$-VAE on the CIFAR-10 dataset using AdamW optimization with a learning rate of $2 \times 10^{-4}$, and investigate the relationship between membership information leakage and latent space regularization. The model was trained with a batch size of 128 and $\beta =$

$\{0.0, 10^{-7}, 10^{-6}, 10^{-5}, 10^{-4}, 10^{-3}, 10^{-2}, 10^{-1}, 0.5, 1.0\}$ to balance reconstruction fidelity with latent space regularization.

For comparison, we considered two variants: (i) a standard $\beta$-VAE trained solely with pixel-level reconstruction and KL divergence losses, and (ii) a $\beta$-VAE augmented with an adversarial discriminator (adversarial $\beta$-VAE) applied between input and reconstructed images. In the latter case, a hinge-loss–based PatchGAN (Isola et al., 2017) discriminator was introduced partway through training, encouraging the reconstructions to be not only faithful but also visually sharper. Across both setups, training was conducted for sufficient epochs to ensure convergence (120 epochs). The latter LDMs are trained based on these frozen VAE encoders. For phase two LDM training, we deliberately train 2048 epochs to ensure the membership information leakage. For training split, we reuse the member split from previous experiments as shown in Table 4. We sample 1000 images from the well-trained LDM checkpoints and calculate the FID to training split.

Specifically, in the adversarial setup, optimization proceeds in two steps per iteration. First, we update the VAE with the usual reconstruction+KL objective and, after a warm-up, add a non-saturating adversarial term (hinge form; equivalent to $-\mathbb{E}[D(\hat{x})]$) gated to turn on only after a chosen global-step threshold (we set this to 2,000 steps in our runs). The adversarial term is weighted adaptively using a VQGAN-style (Esser et al., 2021) gradient-norm ratio on the decoder's last layer and then scaled by a constant factor of 0.8; before the start step, its weight is zero by design. Second, we update the discriminator with a hinge loss on real images $x$ versus reconstructions $\hat{x}$ recomputed in a no-grad branch (to avoid generator gradient leakage), using AdamW with the same learning rate/betas/weight-decay as the VAE optimizer.

## A.7 WHY EXTREMELY EARLY AND LATE $t$ ARE PROBLEMATIC:

Intuitively, $\hat{\varepsilon}_\theta(x,t)|_{t=0}$ is expected to achieve the best performance. Because the noise added to a member at $t = 0$ is expected to be zero (as shown in **case 1**) and non-zero for a non-member. However, in region of low data density, score-matching lacks sufficient evidence to *reliably* estimate the score function (Song & Ermon, 2019). Song & Ermon (2019) argues that training minimizes the expected value of score estimates (here is $\mathbb{E}_{p_{t=0}}\big[\|\hat{\varepsilon}_\theta(x_{t=0},t) - \varepsilon\|^2\big]$), which provides *inaccurate scores* where $p_{t=0}(x)$ is infinitesimal. To be specific, for the input $\big\{x \in \mathcal{R} \mid \{\mathbf{x}_i\}_{i=1}^N \cap \mathcal{R} = \varnothing, \quad \mathcal{R} \subset \mathbb{R}^d\big\}$, $\nabla_x p_{t=0}(x)$ extrapolates erratically. Consequently, for **very early timesteps** ($\sigma_t \ll 1$) the learned field outside the tightly supported member set can plateau or even shrink, nullifying the privacy signal. Increasing $t$ corresponds to *extra Gaussian convolution*, expanding the effective support and regularising the score. Figure 6 plots the average normalised estimator magnitude $\|\hat{\varepsilon}_\theta(x,t)\|$ for $t \in [0, 300]$ on the *member* and *held-out* splits across datasets. Transient fluctuations are confined to the very earliest timesteps ($t \approx 0$), and the maximal gap between the curves typically occurs at early—but not initial—timesteps, indicating that moderately early diffusion steps provide the strongest membership signal.

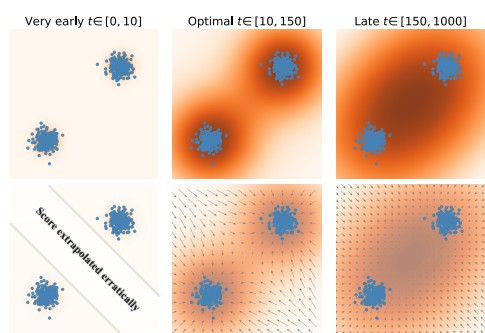

Figure 4: Top: density $p_t(x)$; blue dots are training samples. Bottom: estimated score $\nabla_x \log p_t(x)$. For **very early** $t \in [0, 10]$, the inter-mode region is low-density, so the score extrapolates erratically (shaded band). **Optimal** $t \in [10, 300]$: moderate Gaussian convolution enlarges the support and regularizes the estimator—density bridges the modes and the score points coherently toward them, yielding the strongest separation between members and held-out points. For **late** $t \in [300, 1000]$, $p_t$ approaches an isotropic Gaussian and the score collapses toward the global mean, diminishing membership signal.

Conversely, for **late steps** ($\sigma_t \approx 1$) the forward process approaches an isotropic Gaussian (data information gradually diminish); $p_t$ is nearly homogeneous, so the posterior $\mu_t(x) = q_t(x_0 \sim p_{\text{training}} \mid x)$ collapses to $\mu_t(x) = q_t(x_0 \sim \mathcal{N}(0,I) \mid x)$, which depends on test images and membership information is lost. Figure 4 illustrates the phe-

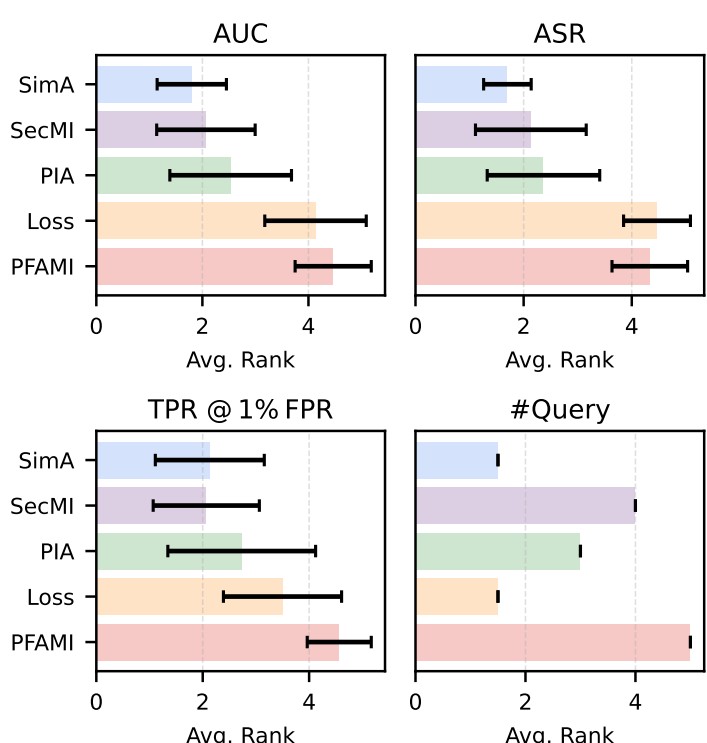

Figure 5: Average ranks ($\pm 1\sigma$) of the five benchmark methods across 15 experiments for four evaluation metrics. Higher values indicate better AUC, ASR, and TPR@1%FPR; lower values indicate fewer #Queries

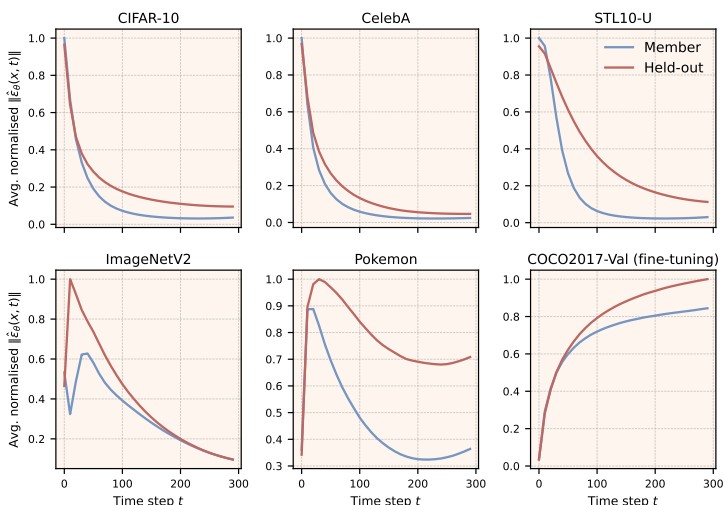

Figure 6: The average normalised estimator magnitude $||\hat{\varepsilon}_\theta(x,t)||$ for $t \in [0, 300]$ on the *member* and *held-out* splits across datasets

nomenon. The *optimal* timestep $t^*$ is therefore dataset-specific and also depends on the noise schedule.

