# OpenReview forum: "Score-based Membership Inference on Diffusion Models"
_ICLR.cc/2026/Conference — ICLR 2026 Conference Withdrawn Submission_

### Official Review · Reviewer_4BiB · 2025-10-29

**Soundness:** 3
**Presentation:** 1
**Contribution:** 2
**Rating:** 4
**Confidence:** 2

**Summary:**

The authors propose an effective membership inference attack against diffusion models. The key idea is to set $\epsilon$ to zero, which enables an effective attack with a single query. They provide theoretical support for the method and show experimentally that it improves TPR. They also find that existing attacks, including the proposed method, perform poorly on latent diffusion models (LDMs).

**Strengths:**

* Proposed a simple yet efficient and effective method.
* Theoretically demonstrated that membership can be distinguished.
* Experimentally showed an improvement in TPR.
* Found that attacks, including the proposed method, perform poorly on latent diffusion models (LDMs).

**Weaknesses:**

* Although the method appears effective, it does not seem novel in terms of either the approach itself or the motivation compared to existing methods.
* The paper only discusses the advantages of setting ε to 0 versus non-zero values at a high level. It would be stronger if the authors provided a theoretical or experimental analysis of specific cases where such differences might arise.
* Most modern diffusion models are based on LDMs, and the fact that the proposed method is not effective compared to other approaches in this setting makes its overall impact appear limited.
* Except for TPR, the improvements in ASR and AUC appear marginal.
* The paper is not well organized. The key idea, setting ε = 0, appears only later in the method section, even though it is central to the approach. This concept should have been introduced earlier, ideally in the introduction and at the beginning of the method section, along with a clear discussion of how it differs from prior methods in the related work. The authors also observed that all attacks, including their proposed one, perform poorly on LDMs, and they analyzed this issue. They further showed that increasing the β value can mitigate such attacks. However, this part feels somewhat disconnected from the main storyline and could potentially be developed into a separate paper.

**Questions:**

* The method is particularly effective on guided diffusion. What is the reason for this?
* ASR and AUC are similar, but the method shows notably better performance only in TPR. What is the reason for this?

---

> ### Author Response · Authors · 2025-11-22
> **Response for Reviewer 4BiB**
>
> We thank you for your time and constructive feedback. Please note that we have a general statement and update to results in a comment on the main thread. Thanks to your comment, our method (with your Monte Carlo suggestion) acheives SoTA on all metrics and all tests. Please see the main comment for details.
>
> There are specific responses to each other point below:
>
> **Reviewer** `4BiB` states that the proposed method is [paraphraising] effective but not novel in approach or motivation.
>   **Our response:** Overall we agree that the broad class of estimators that "look like" ours is not new; our method is the refinement of the naive Loss, and a simplification of multiple other methods (PIA, SecMI). On the otherhand, our exact forumlation, albeit simple, appears to be unique in the literature. Other methods add unnecessary steps or at best confounding terms, while we have by far the simplest formulation. The form of our estimator is derived from our analysis of the ``peaks'' of the distribution defined implicitly by its gradient field, equal to the optimal denoising function.
>
> ---
>
> **Reviewer** `4BiB` suggests that we try a Monte Carlo based approach instead of simply using $\varepsilon=0$.
>   **Our response:** We thank the reviewer for this very helpful comment. We initially avoided this case as we mistakenly believed that an Monte Carlo estimator of the expectation would take a very long time to converge. The Monte Carlo estimator works much better than expected; please see the main response comment, and the updated manuscript.
>
> ---
>
> **Reviewer** `4BiB`, similar to **Reviewer** `ZWwy`, notes that LDM solutions have not been provided.
>   **Our response:** Overall we agree. Though the proposed SimA method with the suggested MC sampling method has SoTA performance, it is still lagging behind non-LDM inversions. This warrants deeper investigation, and remains an open problem.
>
> ---
>
> **Reviewer** `4BiB` notes that ASR and AUC improvements appear marginal.
>   **Our response:** All results have been improved by the use of the Monte Carlo estimator, and we kindly ask the reviewers to reasses the empirical results in light of these new experiments.
>
> ---
>
> **Reviewer** `4BiB` generally does not like the organization of the paper.
>   **Our response:** First, we have a slight refinement to their statement: the key idea is not to set $\varepsilon=0$, but to measure only the norm of the estimator $||\hat\varepsilon(x,t)||$ at test point $x$, not the error/noise residual $|| \hat{\varepsilon}(x+\varepsilon,t) - \varepsilon ||$ or variations upon that measurement (as in SecMI and PIA). While this coincides with setting $\varepsilon=0$, the generalization of the former still only measures the norm of the output of the network, while the latter measures the difference between the noise applied to the test point and the output.
>
>   Otherwise, we agree, and we will introduce the exact form of the estimator in Section 1.
>
> ---
>
> **Reviewer** `4BiB` asks why our method is particularly effective on guided diffusion.
>   **Our response:** We hypothesize that this is due to the dataset, not the particular diffusion model/method. ImageNet has larger dimensionality and many more samples than the other datasets, leading to likely differing characteristics of the optimal denoiser and its geometry around training points.

---

### Official Review · Reviewer_ZWwy · 2025-10-31

**Soundness:** 3
**Presentation:** 2
**Contribution:** 3
**Rating:** 6
**Confidence:** 3

**Summary:**

This paper presents an innovative membership inference attack method called SimA for diffusion models. The core idea is that the norm of the predicted noise at a data point serves as a strong indicator of its membership, based on the theoretical insight that the denoiser output points towards a kernel-weighted local mean of nearby training samples. SimA operates as a ​​single-query attack​​, improving efficiency over multi-query baselines while achieving competitive performance across DDPM, Guided Diffusion, and Latent Diffusion Models. The study reveals that ​​LDMs are inherently more robust​​ to MIAs due to the information bottleneck imposed by their latent autoencoder.

**Strengths:**

1. The paper provides a rigorous derivation linking the predicted noise norm to kernel-weighted local means of training data, offering a unified explanation for score-based MIAs.
2. SimA requires only a single query per sample, reducing computational cost compared to multi-query methods.
3. The study identifies the unexpected robustness of LDMs to MIAs, linking it to VAE regularization hyperparameters and proposing strategies to enhance privacy without sacrificing generative quality.

**Weaknesses:**

1. The paper does not fully explore why VAE-based bottlenecks mitigate MIA beyond empirical correlations with β. A deeper analysis of how exactly the latent space bottleneck disrupts the membership signal would have strengthened this finding.
2. The method is sensitive to the selection of the timestep t and requires manual tuning for different datasets. Is it possible to develop a principled method for selecting the optimal t automatically, thus eliminating the need for this empirical sweeping?

**Questions:**

See the weaknesses.

---

> ### Author Response · Authors · 2025-11-22
> **Response for Reviewer ZWwy**
>
> We thank you for your time and constructive feedback. Please note that we have a general statement and update to results in a comment on the main thread. Thanks to **Reviewer** `4BiB`, our method (with their Monte Carlo suggestion) acheives SoTA on all metrics and all tests. Please see the main comment for details.
>
> There are specific responses to each other point below:
>
> **Reviewer** `ZWwy` notes that the VAE latent space changes to MIA performance are not fully explored.
>   **Our response:** We agree that a deeper analysis is needed. This should be the subject of ongoing research; within the context of our submitted manuscript, the experiments we ourselves are running have become more involved and require math-heavy descriptions that warrant effectively a second submission. At least up to our own [unpublished] work so far, we have no clear conclusions on what exactly changes when using latent codes. We apologize that this has no satisfying answer within this manuscript, as the rabbit hole seems to go much deeper.
>
> ---
>
> **Reviewer** `ZWwy` asks about optimal $t$ selection for the MIA estimators.
>  **Our response:** It may be possible to derive an optimal t criterion, but it surely will be dataset and noise schedule dependent. This is another open problem in MIA: while it would be nice to assume that a validation step is possible, this somewhat contradicts the threat model. A useful example is any standard Diffusion model, trained with noise schedule $\frac{1}{2}$ of the usual. This means that the optimal $t$ should be twice the ``normal'' optimal $t$. Deriving this fact or telling the two models apart without knowing the noise schedules themselves remains unsolved, and no models have made progress or attempts to solve it to our knowledge.

---

### Official Review · Reviewer_RjBy · 2025-11-01

**Soundness:** 2
**Presentation:** 3
**Contribution:** 2
**Rating:** 4
**Confidence:** 3

**Summary:**

This paper focuses on MIAs against DMs. It proposes SimA, a novel single-query MIA framework that leverages the norm of predicted noise vectors, rooted in the insight that the expected denoiser output points to he kernel-weighted local mean of nearby training samples. The paper validate SimA across pixel-space DMs and LDMs on multiple datasets. SimA outperforms multiquery baselines in ASR and AUC. The paper also shows that LDMs exhibits suprisely less vulneralbe to MIAs due to the strong information bottelneck and the need to focus on VAE inversion for MIA on LDMs.

**Strengths:**

1) The paper introduces SimA，a single-query MIA that unifies existig multiquery mothods into a score based framework to avioid relying on Monto Carlo sampling or multi-step queries.
2) The paper gives extensive and rigoruous experimental validation across 11 datasets and 3 major DM classes.
3) The paper identifies a critical problem, that is, LDMs are less vulnerable to MIAs than pixel-space models.
4）The paper addresses a new direction of MIA research from diffusion inversion to VAE inverison.

**Weaknesses:**

1) The paper points out the problem that MIAs exhibit poor performance on LDMs, but it does not pratically resolve this problem.

2) The paper's MIAs are designed under ideal academic assumptions (unrestricted gray-box information access, minimal defenses) that do not align with commercial realities.

**Questions:**

SecMI requires multiple queries to assemble a statistical profile,  but SimA demands richer single-query information (noise vector norms). While SimA appears more efficient than SecMI in query count, both methods fail in real-world commercial settings due to strict black-box constraints.

---

> ### Author Response · Authors · 2025-11-22
> **Response for Reviewer RjBy**
>
> We thank you for your time and constructive feedback. Please note that we have a general statement and update to results in a comment on the main thread. Thanks to **Reviewer** `4BiB`, our method (with their Monte Carlo suggestion) acheives SoTA on all metrics and all tests. Please see the main comment for details.
>
> There are specific responses to each other point below:
>
> **Reviewer** `RjBy` states that while we have identified that membership inference methods have poor performance on LDMs, we do not resolve the problem.
>   **Our response:** Overall we agree. Though the proposed SimA method with the suggested MC sampling method has SoTA performance (see main response), we believe that to truly solve LDM membership inference and model inversion, additional understanding of those models' latent spaces will be necessary, beyond the diffusion process themselves. There may be non-trivial changes in either generative dynamics or training dynamics due to these embeddings.
>
> ---
>
> **Reviewer** `RjBy` states that the grey box problem is idealized, and not realistic to ``commercial realities``.
>   **Our response:** Respectfully, we disagree. While the current paradigm is to produce non-weight-sharing generation-as-a-service products, outside of natural images (e.g., in medical imaging), concerns about privacy and inversion are increasing [1].
>
>   It may also be the case that weight sets might be operated under license, while data are not licensed. This has yet to be litigated, but the possibility of inversion complicates the situation.
>
>   Finally, model inversion may be relevant to other legal questions about tests of inclusion or exclusion of specific training data (namely, data that were not licensed to the model training individuals/groups but still used in a non-fair use manner).
>
>   [1] Unconditional latent diffusion models memorize patient imaging data. *Nature Biomedical Engineering, 2025*
>
> ---
>
> **Reviewer** `RjBy` also has an entry in the Questions portion of the review that appears to be a statement.
>
>   **Our response:** Our apologies, but we don't know how to respond to this, except to say that due to its second term (eq. 18 in our submission), SecMI actually does require the same access as SimA.

---

### Official Review · Reviewer_ejzA · 2025-11-01

**Soundness:** 2
**Presentation:** 2
**Contribution:** 2
**Rating:** 2
**Confidence:** 4

**Summary:**

This paper proposes SimA, a single-query diffusion MIA that leverages the norm of the predicted noise vector (score) from diffusion models. Theoretical analysis shows that the denoiser output points toward a kernel-weighted local mean of training samples, with its norm encoding proximity to the training data. Empirically, SimA achieves strong performance across various diffusion models.

**Strengths:**

1. The paper presents a novel approach to Membership Inference Attack (MIA) by analyzing the norm of the estimated noise.

2. The experiments, particularly the investigation involving the β-VAE, provides interesting results.

**Weaknesses:**

+ Clarity of Core Assumption in SimA​. The paper posits that for member samples, the local mean converges to the training sample itself, implying that at $t=0$, the predicted noise $\epsilon_{\theta}(x_0,t=0)$ (where $x_0$ is the clean training sample) should converge to zero. However, in practice, the output at $t=0$ often resembles Gaussian noise with a norm significantly larger than 0. This discrepancy raises questions about the validity of the local-mean assumption in real-trained models, especially regarding how well the theoretical peak-to-sample alignment is achieved under finite training and network parameter update.

+ Peak Formation for Non-Members​. Figure 1 suggests that members correspond to peaks in the smoothed probability distribution, but it does not sufficiently address whether non-member samples can also form local peaks. If the generative process navigates probability gradients, non-members that are plausible under the data distribution may also lie at or near critical points (local maxima), leading to small noise norms similar to members. What is the structure of the probability distribution near non-members? Why non-members can not form a peak? If non-members also form a peak, its local mean also converges to zero. That means nonmembers and nonmembers can not be distinguished by **the gradient**.

+ ​Distinction Between Data Manifold and Learned Manifold​. I find that the author seems to try to conflate the true data distribution with the distribution learned by the neural network. The argument assumes that the diffusion model’s score function accurately reflects the kernel-smoothed empirical distribution of the training set. In reality, the network only minimizes a training loss and may not perfectly capture the data manifold. The geometric properties of the learned score field could differ significantly. This gap may undermine the core assumption that the local mean around a query point is well-approximated by the model.

+ (Minor) Relationship Between Naive Loss and SimA​ (Line 311-323). The “Naive” baseline (e.g., loss-based MIA) can be viewed as computing $||\epsilon_{\theta}+(-\epsilon)||$, which resembles SimA’s ||\epsilon_{\theta}|| but with an added noise term $-\epsilon$. If the noise term averages out over multiple queries, one might expect the Naive method to converge to SimA’s performance with increased sampling. However, the performance of Naive Loss does not scale with the number of query. How to explain this?

+ (Minor) It is quite strange that PIA achieves the best performance in CIFAR-10 (Table 1), which contrasts with [1] that shows PIA is generally unstable.

[1] Membership inference on text-to-image diffusion models via conditional likelihood discrepancy. Neurips 2024.

**Questions:**

See the weaknesses.

**Details Of Ethics Concerns:**

No ethical statement is provided.

---

> ### Author Response · Authors · 2025-11-22
> **Response for Reviewer ejzA**
>
> We thank you for your time and constructive feedback. Please note that we have a general statement and update to results in a comment on the main thread. Thanks to **Reviewer** `4BiB`, our method (with their Monte Carlo suggestion) acheives SoTA on all metrics and all tests. Please see the main comment for details.
>
> There are specific responses to each other point below:
>
> **Reviewer** `ejzA` raised an issue about the ``Clarity of Core Assumption in SimA`` [listed in weaknesses]:
> **Our response:**
> At $t=0$, behavior is undefined and untrained for $\hat\varepsilon$ networks. Since at least one noise step is always applied, this usually isn't a meaningful input, and might result in the Gaussian noise.
>
> For $t > 0$ but small, $\varepsilon_\theta(x_0, t=\text{small})$ should tend toward zero for the optimal model [1,2,3] at $x_0$ in the training data. This is because $\varepsilon=0$ produces $x_0$ itself, and all other data points are relatively distant unless the infinite data limit is considered.
> In practice, even for heavily parameterized models this inductive bias will induce a systemic gap, and may raise this value above zero.
>
> [1] An analytic theory of creativity in convolutional diffusion models. *ICML 2025*
>
> [2] Towards a mechanistic explanation of diffusion model generalization *ICML 2025*
>
> [3] Locality in Image Diffusion Models Emerges from Data Statistics. *NeurIPS 2025*
>
> ---
>
> **Reviewer** `ejzA` raised an issue about the ``Peak formation at non-member points`` [listed in weaknesses]
> **Our response:**
> First, non-members definitionally have no causal influence on the model training (otherwise they would be training set members). Thus, they cannot cause peaks to form at their position. However, a question immediately arises: are there peaks at non-member positions anyway? This is the topic of our paper, and in Section 3.2 in case 2 we show that generally no, in expectation this does not happen for the optimal denoising model.
>
> ---
>
> **Reviewer** `ejzA` raised an issue about [paraphrasing] not making a distinction between the data manifold and the learned manifold:
> **Our response:**
> The reviewer is right that we do use the optimal denoiser (``data distribution''), not the learned distribution. The optimization/learning-procedure tends toward this optimal denoiser, and we do not consider learning dynamics in this paper, so we cannot guarantee any convergence rate. However, because analytic forms for this optimal denoiser are available, and, in training/validation, we would select the optimal denoiser over other models as it definitionally performs better than every other model, it seems prudent to use it. While it is maybe possible to provide analyses for specific architectures' overfit behaviors if studied quite carefully, we believe that the general cases' desiderata (i.e., the denoiser towards which all parameterized models converge to in the NP limit) is sufficient.
>
> ---
>
> **Reviewer** `ejzA` raised an issue about the similarity between the naive Loss method and the proposed SimA method:
> **Our response:**
> There is one key difference between the Naive method and the proposed SimA method. The loss method measures $\lvert|\hat{\varepsilon}(x+\varepsilon,t) - \varepsilon |\rvert$, which is the noise estimator network $\hat{\varepsilon}$ evaluated at the test point plus $\varepsilon$, and computes its difference with $\varepsilon$. SimA simply computes the norm of $\hat{\varepsilon}$. In our original point estimate, we estimate this for $\varepsilon=0$, which is the mode/maximum density of the noise. In our updated experiments, we expand this to Monte Carlo sampling on $\varepsilon$.
>
> This does _not_ average out over many samples. We can derive it for the $\ell_2$ norm, though the difference has different (yet still non-zero) values for different norms
>
> $$ Loss = \lvert| \hat{\varepsilon}(x+\varepsilon,t) - \varepsilon \rvert|_2^2  $$
>
> $$ =\lvert|\hat{\varepsilon}(x+\varepsilon,t) \rvert|_2^2 + \lvert| \varepsilon \rvert|_2^2 - 2 \hat{\varepsilon}(x+\varepsilon,t)^T \varepsilon $$
>
> $$ = SimA + \sigma^2 - 2 \sigma^2 \sum_{i} \frac{\partial }{\partial x_i} [\hat{\varepsilon}(x+\varepsilon,t)]_i $$
>
> Use stein's lemma for the last step. SimA is related to Loss, clearly (all the methods have similar forms in this class), they're not equivalent.
>
> ---
>
> **Reviewer** `ejzA` stated that it was surprising that PIA had stable performance.
> **Our response:**
> For PIA implementation, we reused the codes from their GitHub repository. This code is found in https://github.com/kong13661/PIA, and we made no meaningful modifications, only adapting it to the scripts running our experiments.

---

### Author Response · Authors · 2025-11-22
**Main Response: Additional LDM Experiments and a Monte Carlo Variant of SimA Provide SoTA Performance. We kindly ask the reviewers, ACs/SACs, and PCs to read through these experiments, as they significantly change the results of our paper**

We thank all reviewers for their time and constructive feedback.

Reviewer 4BiB's second point about selection of $\varepsilon=0$ prompted us to implement and run a Monte Carlo version, using $\varepsilon \sim \mathcal{N}$. We had previously avoided this, thinking that such a high dimensional Monte Carlo would perform poorly until a very large number of samples. Specifically, for a test image $x$, $\lVert\hat\varepsilon_{\theta}(x,t)\rVert$ (**SimA**) is derived from $\mathbb{E}{\varepsilon\sim\mathcal N(0,I)}[\lVert\hat\varepsilon_{\theta}(x_t,t)\rVert]$ (**Monte Carlo variant of SimA**) at $\varepsilon=0$ (point estimator), also described in the first figure.

**Instead, the Monte Carlo variant of our proposed method provides SoTA performance within 30 samples, often within 10**. For sufficient ($n \in [10,30]$) samples, it improves all performance metrics on all datasets, some by more than 30 points (substantially increases TPR@1%FPR). We detail the exact changes in the next section, and the resulting outcome changes.

This significantly changes the results of our paper. While we understand that, for conference submissions, significant changes are usually discouraged during rebuttal, we feel that it is necessary to report these new results. We have accordingly modified our submission, and we ask the reviewers to kindly read through the changes. We apologize for the additional workload.

We have read through the ICLR policies, and we feel that this is in line with ICLR rebuttal policy, but should the AC/SAC or PC chairs feel otherwise, and feel that we have changed the submission too much from its original state, we will withdraw the submission.

### Updated Experiment Details

For latent diffusion models (LDMs), beyond the experiments we provided in the appendix, we conduct additional experiments on additional datasets. **Table 1 and Table 2 in the next comment report the results.** These include a Monte Carlo variant of our proposed estimator, with the number of Monte Carlo samples listed as (#mc) next to the method. We now have **SimA**, **SimA (#mc=10)**, and **SimA (#mc=30)**.

**Dataset Details**: For Imagenet-1k, we used 100K (#100 for each class) as the training set. We used the Imagenet-1k validation set (instead of ImageNetv2) as the held-out set. For the first table, we mimic the network architecture of Stable Diffusion (not 100% the same due to its text-conditioning). A wide range of image resolution ($32^2$ for CIFAR-10, $64^2$ for CelebA, $256^2$ for ImageNet-1K, $512^2$ for stable diffusion experiments) has been evaluated. For the stable diffusion experiments, we have retrained all models; we lost the previous checkpoints in a disk failure. This resulted in slightly different numbers, but overall qualitatively similar outcomes (same rankings, same relative regimes).

**Results  (See next comment)**: From the first tables (additional experiments), SimA and its MC variants consistently outperform all other methods, in all datasets, for all measured performance metrics. As the number of Monte Carlo samples increase, the attack performance increases as well. From the second table, MC of SimA still outperforms.

We will update the submission accordingly to reflect these changes. We thank the reviewers for their time.

---

> ### Author Response · Authors · 2025-11-22
> **Experimental Results**
>
> **Table 1**: Attack performance of four baseline methods used to evaluate memorization of **LDMs**. Apart from SimA(\#mc), best results of in **bold**. The gray area denotes the Monte Carlo variant of SimA. \#mc denotes the number of Monte Carlo samples. **At \#mc=10, SimA advances the state-of-the-art across all reported metrics**.
>
> | Method                | \#Query$\downarrow$ | CIFAR-10 AUC↑ | CIFAR-10 ASR↑ | CIFAR-10 TPR@1\%FPR↑ | CelebA AUC↑ | CelebA ASR↑ | CelebA TPR@1\%FPR↑ | ImageNet AUC↑ | ImageNet ASR↑ | ImageNet TPR@1\%FPR↑ |
> |-----------------------|---------------------|---------------|---------------|----------------------|-------------|-------------|---------------------|---------------|----------------|------------------------|
> | Loss                  | 1                   | 73.28         | 67.48         | 7.86                 | 68.65       | 63.47       | 6.39                | 67.49         | 63.16          | **4.09**               |
> | SecMI                 | 12                  | 87.42         | 80.12         | 19.11                | 83.09       | 75.85       | 10.53               | 68.21         | 63.44          | 3.71                   |
> | PIA                   | 2                   | 85.73         | 78.19         | 15.90                | 83.36       | 75.87       | 9.41                | 66.09         | 62.52          | 2.95                   |
> | SimA ($\varepsilon=0$) | 1                  | **89.10**     | **81.63**     | **19.88**            | **84.66**   | **77.04**   | **11.09**           | **69.62**     | **64.92**      | 3.87                   |
> | SimA (\#mc=10)        | 10                  | **90.68**     | **82.71**     | **39.86**            | **91.55**   | **83.56**   | **39.76**           | **71.13**     | **65.95**      | **7.44**               |
>
> ---
>
> **Table 2**: Attack performance of four baseline methods used to evaluate memorization of **Stable Diffusion**. Apart from SimA(\#mc), best results in **bold**. The gray area is the Monte Carlo variant of SimA. \#mc denotes the number of Monte Carlo samples. **Notably, at \#mc=10 and 30, SimA advances the state-of-the-art across all reported metrics**.
>
> | Method                 | \#Query$\downarrow$ | Pokémon AUC↑ | Pokémon ASR↑ | Pokémon TPR@1\%FPR↑ | MS-COCO AUC↑ | MS-COCO ASR↑ | MS-COCO TPR@1\%FPR↑ | Flickr AUC↑ | Flickr ASR↑ | Flickr TPR@1\%FPR↑ |
> |------------------------|---------------------|-------------|-------------|----------------------|--------------|--------------|----------------------|-------------|-------------|---------------------|
> | Loss                   | 1                   | 92.03       | 85.47       | **40.77**            | 83.26        | 75.80        | 13.48                | 63.13       | 59.74       | 2.35                |
> | SecMI                  | 12                  | 88.12       | 81.04       | 29.02                | 91.35        | 84.00        | **39.16**            | **71.81**   | **66.54**   | **5.29**            |
> | PIA                    | 2                   | **94.93**   | **90.52**   | 32.85                | 92.51        | 85.72        | 24.76                | 68.88       | 64.61       | 2.67                |
> | SimA ($\varepsilon=0$) | 1                   | 93.50       | 87.87       | 20.38                | **93.71**    | **87.34**    | 29.80                | 70.04       | 65.96       | 2.59                |
> | SimA (\#mc=10)         | 10                  | **96.75**   | **91.47**   | **63.79**            | **93.21**    | **85.76**    | **45.36**            | **70.26**       | **65.65**       | **4.47**                |
> | SimA (\#mc=30)         | 30                  | **97.01**   | **92.31**   | **70.50**            | **94.24**    | **86.66**    | **52.48**            | **72.23**   | **66.85**   | **6.33**            |

---

### Note · Authors · 2026-04-14

I have read and agree with the venue's withdrawal policy on behalf of myself and my co-authors.

---

### Meta-Review · Area_Chair_XPVb · 2026-01-06

**Summary:**

This paper studies membership inference attacks on diffusion models and proposes SimA, a single-query attack based on the norm of the predicted noise.
Reviewers agreed that the topic is relevant and timely, and that the proposed attack is simple and efficient. Several reviewers found the empirical results on pixel-space diffusion models to be strong, particularly after the rebuttal introduced a Monte Carlo variant that significantly improved attack performance. However, it may request another round of review since this is a new method proposed. Also, concerns were raised about the strength of the theoretical assumptions, the novelty of the approach relative to prior score- or loss-based MIAs, and the limited effectiveness of all evaluated attacks on latent diffusion models. Additional concerns included clarity and organization, reliance on an idealized gray-box threat model, sensitivity to timestep selection, and the lack of a clear explanation for why latent autoencoders appear to reduce membership leakage.

The rebuttal added substantial experimental results and clarified several technical points. The paper would be beneficial for another round of review given the updated methods and experimental results.

**Reviewer Concerns:**

### Concerns addressed by the rebuttal:

- The relationship between SimA and prior MIA methods was clarified, including why the proposed method is not equivalent to existing loss-based or score-based attacks.

- Questions about the behavior of the method at very small timesteps were addressed by clarifying the practical interpretation of the theoretical analysis.

- Reviewer questions regarding baseline instability and implementation details were clarified.

- The authors acknowledged issues with organization and indicated plans to restructure the paper to make the core contribution clearer.

### Concerns that remain outstanding:

- The newly added method significantly improve the experimental results of the paper, but may need another round of review. Otherwise, the performance on LDM is still relatively poor.

- The assumed gray-box threat model may limit practical relevance in real deployment settings.

- The contribution is viewed by some reviewers as incremental rather than fundamentally new.

- The need to tune the diffusion timestep for effective attack performance remains an open issue.

**Reviewer Scores:**

Reviewer ejzA: Initially 2. I think the authors answered the questions well and the review may increase the socre.

Reviewer RjBy: Initially 4. Concerns were partially addressed, the review may keep the rating.

Reviewer ZWwy: Initially 6. Likely to remain at the same score.

Reviewer 4BiB: Initially 4 and contributed feedback that led to major improvements. Likely to increase the score.

---

### Decision · Program_Chairs · 2026-01-26

Reject